# Nonclinical Evaluation of Single-Mutant *E. coli* Asparaginases Obtained by Double-Mutant Deconvolution: Improving Toxicological, Immune and Inflammatory Responses

**DOI:** 10.3390/ijms25116008

**Published:** 2024-05-30

**Authors:** Grace Ruiz-Lara, Tales A. Costa-Silva, Jorge Javier Muso-Cachumba, Johanna Cevallos Espinel, Marina Gabriel Fontes, Mitla Garcia-Maya, Khondaker Miraz Rahman, Carlota de Oliveira Rangel-Yagui, Gisele Monteiro

**Affiliations:** 1Department of Biochemical and Pharmaceutical Technology, School of Pharmaceutical Science, University of São Paulo, São Paulo 05508-000, SP, Brazil; gvruizlara@usp.br (G.R.-L.); jjaviermusoc@usp.br (J.J.M.-C.); marina.fontes@usp.br (M.G.F.); corangel@usp.br (C.d.O.R.-Y.); 2Center for Natural and Human Sciences, Federal University of ABC, Santo André 09210-580, SP, Brazil; tales.costa@ufabc.edu.br; 3Eugenio Espejo Specialty Hospital, Quito 170136, Ecuador; johannacevallosespinel@gmail.com; 4Randall Division of Cell and Molecular Biophysics, King’s College London, London SE1 1UL, UK; mitla.garcia@kcl.ac.uk; 5Institute of Pharmaceutical Science, King’s College London, London SE1 9NH, UK; k.miraz.rahman@kcl.ac.uk

**Keywords:** *E. coli* asparaginase, bioengineer, leukaemia, nonclinical assays

## Abstract

Acute lymphoblastic leukaemia is currently treated with bacterial L-asparaginase; however, its side effects raise the need for the development of improved and efficient novel enzymes. Previously, we obtained low anti-asparaginase antibody production and high serum enzyme half-life in mice treated with the P40S/S206C mutant; however, its specific activity was significantly reduced. Thus, our aim was to test single mutants, S206C and P40S, through in vitro and in vivo assays. Our results showed that the drop in specific activity was caused by P40S substitution. In addition, our single mutants were highly stable in biological environment simulation, unlike the double-mutant P40S/S206C. The in vitro cell viability assay demonstrated that mutant enzymes have a higher cytotoxic effect than WT on T-cell-derived ALL and on some solid cancer cell lines. The in vivo assays were performed in mice to identify toxicological effects, to evoke immunological responses and to study the enzymes’ pharmacokinetics. From these tests, none of the enzymes was toxic; however, S206C elicited lower physiological changes and immune/allergenic responses. In relation to the pharmacokinetic profile, S206C exhibited twofold higher activity than WT and P40S two hours after injection. In conclusion, we present bioengineered *E. coli* asparaginases with high specific enzyme activity and fewer side effects.

## 1. Introduction

Pharmacological, toxicological and pharmacokinetic preclinical studies are essential for developing any biopharmaceutical and must follow the guidelines of the International Conference on Harmonization of Technical Requirements for Registration of Pharmaceuticals for Human Use (ICH) Document S6, title “S6 Preclinical Safety Evaluation of Biotechnology-Derived Pharmaceuticals”, from 1997 [1]. In 2010, a specific guidance called “Nonclinical Evaluation for Anticancer Pharmaceuticals” was released [2]. According to the document, a successful nonclinical experiment should consider important details such as pharmacological properties of the drug, non-toxic doses for future extrapolation to humans and a toxicological profile to identify damage to specific organs and reversible/irreversible effects. In addition, in 2012, an S6 Addendum to Preclinical Safety Evaluation of Biotechnology-Derived Pharmaceuticals was published with complementary, useful and updated recommendations for in vivo assays.

In this context, well-detailed nonclinical trials for a novel asparaginase biopharmaceutical are critical for future successful clinical trials. Currently, the L-asparaginase enzyme from bacteria, commonly known as ASNase, is widely used worldwide to treat blood cancer, such as acute lymphoblastic leukaemia (ALL) in children and adolescents, where it has been highly successful. The first FDA-approved enzymes were ASNases from *Escherichia coli* (EcA) and *Erwinia chrysanthemi* (ErA) in 1978 and 2011, respectively. However, side effects related to their use, mainly in adult patients, such as neurotoxicity, pancreatitis, hypersensitivity or even anaphylactic shock [3], necessitate improved versions of this enzyme. In this sense, preclinical toxicological and immune assays are crucial to proceed with the clinical trials. Studies with different ASNases on several biomodels have been reported in the literature under different treatment schemes [4,5,6,7,8]. Herein, we present a detailed study to complement a previously reported study with double-mutant P40S/S206C [9] and its separated mutations (P40S and S206C) to better characterise their specific effect on animals.

Several studies have explored better catalytic properties of ASNase to overcome side effects, including the prospection of new ASNase sources or the development of biobetters [4,10,11,12]. Protein bioengineering has emerged as an excellent tool that offers researchers the possibility to create new proteins/enzymes with desired profiles, such as improved shelf-life and serum half-life stability, better substrate affinity/selectivity and low immunogenicity [13,14,15]. In this context, our research group has been focused on protein bioengineering techniques to create ASNase biobetters. Applying this approach, several *E. coli* ASNase clones were developed and, after in vitro and in vivo assays, it was suggested that the P40S/S206C double mutant could be an interesting option for ALL treatment as it presented improved plasma half-life and induced low anti-ASNase antibody formation [9]. However, the P40S/S206C double-mutant enzyme showed a significant loss of specific activity to asparagine hydrolysis and exhibited high relative glutaminase activity. These features prompted a deeper investigation of the double-mutant ASNase P40S/S206C. Therefore, we applied a deconvolution tool to this double mutant [16], exploring the individual contributions of each mutation to its improved resistance, stability and immune system evasion.

Thus, our goals with this study were to characterise and evaluate the biological in vitro activity of single-point mutation ASNase P40S and S206C in comparison with the double mutant from Rodrigues’ work (P40S/S206C). Additionally, we aimed to determine whether the encouraging results stemmed from the double mutation interaction or if the specific activity could be restored by a single-point mutation, while preserving increased serum half-life and low antibody production. Furthermore, published studies of cytotoxicity assays with mutant ASNases on solid tumours are limited. Hence, we evaluated the potential effect of these mutants on immortalised adherent cancer cell viability. Finally, to conduct a detailed in vivo assay with a multiple-dose scheme, we conducted toxicological analyses and measured the hypersensitivity and inflammatory responses of these proteoforms. These nonclinical studies are essential for research groups to design and implement a safe and effective treatment scheme for clinical studies in humans.

Our results indicated that the P40S mutation was detrimental to enzyme activity, whilst the isolated S206C mutant kept its ability to hydrolyse asparagine similarly to the WT enzyme. Nonetheless, both single mutants were effectively cytotoxic against ALL cell lines. Interestingly, the single mutants presented different behaviours in terms of effectiveness against solid tumour cell lines, induction of antibody titres and physiological alterations related to biopharmaceutical toxicity. Overall, the S206C mutant appears to be the best option, but each proteoform displays distinctive features and may be more suitable for different situations and/or patients. This reinforces the importance of generating new options for ASNase variants for future personalised therapeutic applications.

## 2. Results

### 2.1. Biochemical Characterization of ASNases

The asparaginase gene from *E. coli* (*ansB* gene), as well as its mutants, inserted on vector pET15b, were properly expressed. The proteoforms were successfully purified with a purity percentage above 95% deemed acceptable to proceed with the experiments (Appendix A).

The results for asparaginase (ASNase activity) and glutaminase (GLNase activity) activities are summarised in Table 1 (Appendix A). It was observed that ASNase-specific activity was higher for the native enzyme, herein called wild type (WT), and S206C, at 76.2 and 74.9 U/mg, respectively, than for P40S (60.8 U/mg) and the double-mutant P40S/S206C (52.9 U/mg) proteoforms. As previously mentioned, the P40S/S206C protein was also developed in our laboratory, and, as reported by Rodrigues (2020), it lost around 30% to 40% of the activity compared to the WT enzyme [9]. This may be an issue in leukaemia treatment since more enzyme mass would be required to achieve the therapeutic effect.

Regarding GLNase activity, which is reported to be the main cause of adverse effects from asparaginase therapy [12,17], a decrease was observed for all mutant enzymes (less than 2 U/mg), with the lowest value of 1.3 U/mg observed for the P40S protein (Table 1). All the other mutants presented similar results. However, when correlating with ASNase activity, the double-mutant P40S/S206C enzyme has the higher percentage of intrinsic relative GLNase activity (3.3%), followed by WT, S206C and P40S proteins, with 2.4%, 2.3% and 2.1%, respectively (Table 1).

To simulate a physiological environment, the activity of the WT, S206C, P40S and P40S/S206C enzymes was tested in the presence of human serum (HS) as a measure of in vitro stability. This experiment was conducted before the nonclinical assay to select the enzymes that would be evaluated in vivo for toxicity and immune hypersensitivity, as outlined in our proposed scheme in Section 4.4.2. Therefore, ASNase activity was periodically quantified during 96 h of incubation with HS and statistical differences were observed from 72 to 96 h. As shown in Table 2, the activity of the P40S/S206C enzyme considerably decreased in both PBS 1x (80% lower) and HS (56% lower) after 96 h; its activity started to decrease after 72 h of incubation (Appendix A). In contrast, the WT, S206C and P40S enzymes only lost activity in PBS 1x after 96 h. Also, from Table 2, the highest stability on HS was observed for the S206C enzyme, followed by P40S, WT and P40S/S206C, with residual activities of 83%, 76%, 70% and 44%, respectively. This means that the double mutant conserved only half of the ASNase activity of the single mutant S206C; indeed, even in PBS 1x, the activity was twofold higher for the S206C, WT and P40S proteins compared to the P40S/S206C residual activity.

Our results show that the single mutants S206C and P40S conserved high catalytic activity, similarly to WT, while also maintaining high stability in human serum.

### 2.2. Kinetic Parameters of ASNases

Figure 1 shows the kinetic profiles of commercial Leuginase^®^, WT and single-mutant ASNases, all of which display a non-allosteric Michaelis–Menten behaviour. The kinetic parameters are summarised in Table 3, and from these results, we concluded that the WT and S206C enzymes had lower K_M_ than commercial Leuginase^®^ and P40S proteoform. Nonetheless, V_max_ was similar for all the ASNases tested.

### 2.3. Nonclinical Studies of ASNases

#### 2.3.1. In Vitro Assays

Table 4 indicates the cytotoxicity results of WT ASNase and its mutant proteoforms against T cell (MOLT-4) and B cell precursor (REH) ALL. The P40S enzyme has a significantly higher anti-tumour effect on T cells (MOLT-4) than WT. For REH cells, IC_50_ values were similar for WT and mutants.

From the MTT assay results, shown in Table 5, relative to the WT protein, the S206C enzyme had a significantly lower IC_50_ on Caco-2, PANC-1 and U-87 MG cell lines, while the P40S enzyme is more cytotoxic to PANC-1 and U-87 MG. However, on an ovarian cancer cell line (SK-OV-3), the S206C mutant was less effective and consequently had higher IC_50_ than the WT and P40S ASNases. On the other hand, the P40S enzyme had the highest cytotoxic effect on glioblastoma cells (U-87 MG) with significantly lower IC_50_ compared to WT and S206C. Our results confirm that the mutant enzymes are as active as WT on these solid cancer cell lines; indeed, they even have higher cytotoxic effects against some cell lines despite their mutations. On the other hand, for the breast cancer line MDA-MB-231, the P40S ASNase has lower cytotoxic potential. Taken together, the results highlight the importance of developing diversity variants of ASNase since specific cytotoxic effects are observed for enzymes with the same biochemical function—depleting asparagine but with different consequences for tumour cell line viability.

#### 2.3.2. In Vivo Assays

Following the guidelines of S6 for anticancer pharmaceuticals, and after characterising the enzymes, testing their catalytic potential and stability, and confirming their anti-tumour activity on leukaemic and some solid cancer cells in vitro, we performed the nonclinical assay in mice. The results of the treatment scheme described for toxicity analysis are presented below, as well as the immune response and pharmacokinetic profile (Section 4.4.2).

For toxicity tests, the animals’ health must be monitored throughout the experiments to avoid unnecessary suffering. Thus, body weight was considered an indicator of wellness, which showed an increasing pattern in all groups throughout the experiment with no significant differences among them (Figure 2A).

Regarding the body temperature, no variations from normal temperature were detected on days 0 and 14 after single doses of 1050 U/Kg of ASNase. However, on day 23, after the challenge dose (fivefold higher), a decrease in body temperature was observed in all groups except for the control group (treated solely with buffer). Mice treated with the P40S and S206C enzymes experienced the greatest drop in body temperature around 30 min after the injection, while it occurred around 45 min for the group treated with the WT enzyme. For all groups, body temperature was recovered within 2 h after treatment. Indeed, only minor significant differences were found between body temperatures of the control group against all enzyme-treated groups (Figure 2B).

Blood biometry was evaluated after the challenge dose on day 23. For all groups—control and ASNase-treated with WT, S206C and P40S, ordinary counts of red and white blood cells were detected, except for haematocrit and granulocytes. Although they are within normal ranges, the highest values observed for both parameters resulted from the animal group treated with the P40S enzyme, followed by groups treated with the S206C and WT enzymes (Figure 3A,B). On the other hand, platelet concentration analysis (Figure 3C) exhibited significant differences between the control and all enzyme-treated groups, but also between WT and S206C. The results showed less damage in platelet count when the mutant proteoform S206C was applied in comparison to other ASNases.

In addition, the average weight of some organs from all mice is presented in Table 6. Animals treated with enzymes, both WT and mutants, had significantly reduced pancreas mass and, except for the S206C protein-treated group, increased liver mass when compared with the control group. Also, histopathological results indicated that the morphology and architecture of organs were preserved in the animals of the control group within normal parameters. On the other hand, all groups treated with ASNases presented mild secondary histological alterations. The most frequent findings were microvesicular steatosis of the liver, acute tubular necrosis of the kidneys and mild intercellular oedema of the heart; other tissues were conserved, and no damage was found. Pictures from the WT, S206C and P40S groups are shown in Appendix A, and no major differences were found among WT and mutant enzyme-exposed groups.

The expression of immunoglobulins IgG and IgE anti-ASNase due to enzyme administration, commonly related to hypersensitivity, was quantified, as well as platelet-activating factor (PAF) and monocyte chemoattractant protein (MCP-4), mediators of several leukocyte functions, inflammation, sepsis and anaphylaxis [18,19]. S206C mutant enzyme induced a lesser humoral response throughout the experiment since IgG and IgE levels were lower than WT and P40S-treated groups, especially in the fourth week. Indeed, the P40S-enzyme-treated animal group exhibited the highest IgE titre among of all groups since the second week, while IgG levels increased in the fourth week (Figure 4). Regarding PAF and MCP-4, animals injected with the P40S enzyme showed the lowest concentration of both compounds, with the highest values observed in mice treated with WT (PAF) and S206C (MCP-4) enzymes, respectively (Figure 5).

During this assay, body weight and temperature after each injection of ASNase were also monitored. As expected, all animals gained weight normally throughout the entire experiment, and no variations in temperature were observed after each injection. At the end of the assay, no alterations were detected on blood biometry. All these results highlight that the treatments had a low impact on the animals’ health.

Finally, asparaginase activity was measured in mice plasma at 2, 6, 12 and 24 h after a single dose injection of 1050 U/Kg of each ASNase, WT and mutants (as described in Section 4.4.2). As expected, the maximum enzyme activity was detected 2 h after the injection in all cases, and it decreased thereafter, reaching its minimum after 24 h. No significant differences were found between all enzymes from 6 h until 24 h, except at 2 h post-injection when S206C residual plasma activity was significantly higher than WT and P40S (Figure 6).

## 3. Discussion

Considering the ALL-BFM-IC 2009 protocol, which recommends an ASNase dose of 5000 U/m^2^ every third day on days 12, 15, 18, 21, 24, 27, 30 and 33 (eight doses) during the induction phase, an equivalent scheme in mice should involve around five doses within a 24-day treatment period. Mice used in these experiments were aged 6 to 8 weeks, with an average weight of 20 g and an estimated body surface area of 0.007 m^2^ [20], which correlates with a 13-year-old human adolescent [21]. Thus, the animal equivalent dose in mice should be around 1900 U/Kg [22]. However, we decided to use a three-concentrated-doses scheme, considering that our enzyme formulations did not have stabilisers and osmolytes. Therefore, doses of 1050 and 5250 U/Kg were administered to the animals on days 0, 14 and 23 to investigate whether these novel mutants evoke toxic responses in healthy and immunocompetent animals. In this context, discomfort or pain, animal weight, and corporal temperature were analysed, as well as the organ’s weight, morphology, and histology [5,23,24]. Animal weight loss is associated with toxicity or discomfort since it can be caused by a lack of appetite due to pain/allergies. However, it can also result from behavioural factors such as stress, lack of adaptation or fear, metabolic changes such as malabsorption syndrome or increased physical activity [25]. Indeed, to guarantee animal welfare, in the case of losses equal to or greater than 20% of body weight and lack of water consumption, the humane endpoint should be performed [25,26]. Our results did not demonstrate progressive chronic diseases or acute toxicity due to the enzyme administration since, until the end of the experiment, significant animal weight loss was not observed in any of the tested groups; they all grew normally.

Regarding the analysis of body temperature over two hours after each enzymatic formulation injection (days 0, 14 and 23), the first two doses were harmless, and no variation was observed. In contrast, the temperature decreased at the last challenge dose within each group. In general terms, the maximum loss of body temperature was 1.9 °C for the WT group, 2.8 °C for the S206C group and 2.6 °C for the P40S group. Some authors consider a drop in body temperature as the first symptom of a hypersensitivity reaction [5,24]. In that context, except for the control group, all animals experienced a drop in body temperature within the first hour after injection; however, temperature soon recovered. This indicates that even at higher doses than those recommended by the ALL-BFM-IC 2009 protocol, these mutant enzymes were not lethal. This could be considered as an induced hypothermia for an initial anaesthetic effect due to the introduction of the “foreign body”, the enzymatic solution, rather than a toxicity effect [27]. Under standard conditions, mice have normal temperatures between 35.5 and 37.5 °C with mild hypothermia ranging from 32 to 35 °C [28]. Hence, temperature values recorded after the last dose indicate a transient mild hypothermia for about 2 h. Like weight loss, a drop in body temperature can be considered a symptom of animal discomfort; however, it can also be related to a compensation and protection system [28]. Hypothermia has neuroprotective effects in ischemia or encephalopathy in animal models and humans [29,30,31,32,33]. In the case of ischemia, it can reduce the cerebral metabolic rate and preserve energy [34], decrease the release of free radicals [35], minimise the formation of cerebral oedema and stabilise membranes [36], and also reduce the rate of apoptosis [30,37,38].

As for the blood biometry results, the decrease observed in platelet count may be explained by anti-asparaginase antibody production. Bougie and collaborators (2010) demonstrated that transient thrombocytopenia can be induced by the presence of anti-drug antibodies. Therefore, anti-asparaginase antibodies may cause thrombocytopenia, with the worst scenario observed in the WT group followed by the P40S enzyme, although no animals suffered from spontaneous bleeding. Morowski and collaborators (2013) observed that reductions of 70 to 80% of the normal number of platelets do not prevent the correct formation of thrombi in the case of injuries or ruptures of veins or small arterioles and that even small amounts of platelets (approximately 3%) are effective in maintaining homeostasis without the presence of spontaneous bleeding [39]. Based on this, all animals were capable of living without any bleeding complications; furthermore, the decrease in platelet count is transitory due to enzyme administration. Our results show that the S206C proteoform has the lowest impact on platelet count, which can be an advantage for patients with ALL. Additionally, it is important to mention that higher haematocrit (HCT) percentage in P40S-treated animals may be due to dehydration or shock caused by the mutant itself, unlike the WT or S206C enzyme effect. Moreover, the elevated concentration of granulocytes observed upon P40S ASNase administration may be correlated with enzyme digestion and consequently antigen presentation to the immune system, in line with the higher expression of IgG and IgE observed.

Michael and colleagues gathered regulatory guidelines and data from several pharmaceutical companies located in Europe, North America and Japan regarding toxicity in rodents [40]. Accordingly, the organs analysed in this work are considered relevant to evaluating drug toxicity. The liver, for instance, which reflects physiologic and metabolic distress, is suitable for identifying hepatocellular hypertrophy that corresponds with histopathological changes, peroxisome proliferation and lipidosis. Also, the liver exhibited little animal-to-animal variation and has a primordial function as it is the animal’s primary detoxification organ. On the other hand, kidneys are frequently a target of toxic compounds and could reflect acute injuries. The thymus and spleen are essential as they serve as important indicators of immune toxicities, physiological discomfort and stress, and, in addition, histopathological changes may correlate accurately with organ weight alterations. Moreover, the heart could be valuable due to its limited inter-animal variability and its sensitivity to identifying toxicity [40]. In this context, the results of these experiments indicate minor hepatotoxicity and microvesicular steatosis, which can result in increased fat accumulation in liver cells in all enzyme-treated groups and increased organ size, except for the S206C group. This may be caused by an impairment of liver metabolic functions, including defective fatty acid oxidation, enhanced lipogenesis, irregular triglyceride secretion or increased absorptions of fatty acids from the diet [41,42]. Hepatic failure, like platelet decrease, may be related to anti-asparaginase antibodies such as IgG and IgE, which may also form and accumulate immune complexes and could be responsible for the increased organ size in WT- and P40S-enzyme-treated groups, unlike the S206C-exposed group. It is possible that the animals treated with the S206C enzyme did not have an enlarged organ due to less recognition of the mutant enzyme by host antibodies, as also seen for the double-mutant P40S/S206C [9]. This could mean less accumulation of immune complexes from the blood to hepatocytes through the portal vein and, therefore, the liver was not impaired. Instead, S206C mutation could give the enzyme the ability to escape from hepatocytes and liver macrophages/monocytes from degradation/clearance, circumventing its binding to pattern recognition receptors (PRRs) as a pathogen-associated molecular pattern (PAMP) or damage-associated molecular pattern (DAMP). In this sense, avoiding phagocytosis and degradation by lysosomal proteases may prevent S206C liver bioaccumulation or an exacerbated immune response while maintaining its blood asparaginase activity higher than the other enzymes after injection [43,44].

Rathod et al. (2019) reported that ASNase binds mainly to basophils and B cells and to a lesser extent to neutrophils and macrophages, following two sensitisation doses of 10 mg of *E. coli* ASNase and a challenge dose of 100 mg on day 24 in 8-week-old female C57BL/6 mice [24]. This scheme of treatment was used to induce humoral hypersensitivity through immune complex sensitisation and a re-exposure to the antigen with a challenge dose, similar to our assay. The authors also found that ASNase binding to immune cells could occur either freely (cell-associated IgE) or through anti-ASNase IgG and IgE immune complexes, mainly to basophils, which express both high-affinity IgE receptor FcεRI and low-affinity IgG receptor FcγRIII. Thus, our results indicate that anti-ASNase IgG-mediated hypersensitivity after the challenge dose was significantly higher in WT- and P40S-enzyme-treated groups on the fourth week, unlike the S206C enzyme, which induces the lowest titre of both anti-ASNase IgG and IgE throughout the experiment. Indeed, IgE concentration in mice in WT- and S206C-enzyme-treated groups slightly increased from the second to the fourth week, unlike P40S-exposed animals, which effectively produced a high and increasing titre of anti-ASNase IgE in both measurements.

Hypersensitivity reactions developed to ASNase are mainly related to the induction of CD4+ T cells rather than B cells [24]. Therefore, MCP-4 chemokine was also quantified as it preferentially attracts T cells, monocytes and eosinophils. In addition, it is considered homologous to human chymase because both share high substrate specificity, similar tissue distribution and functional serglycin-binding properties [45,46]. Our results showed the highest MCP-4 increase in the S206C group. As previously reported by Medjene et al. (2020), this observed chymase increase can be protective since it had a potent anti-inflammatory effect in mice with renal ischemia–reperfusion injury and bacterial infection by controlling neutrophil extravasation activation and, consequently, limiting their contribution to the possible associated injuries toward the pathogen/antigen response [45,47]. MCP-4 was recently inversely correlated with IgE levels [45]. Indeed, the P40S-enzyme-treated group had the highest titre of anti-ASNase IgE and the lowest MCP-4 concentration. The same correlation is true for mice treated with the S206C enzyme—high MCP-4 concentration and low IgE titre. Relative to PAF expression, anti-ASNase IgE induces the release of PAF from granulocytes upon re-exposure to the enzyme on day 23, and anaphylaxis reactions may occur. In contrast, if a PAF receptor antagonist is used, hypersensitive reactions are considerably reduced [24]. In this context, significantly increased concentrations of PAF were found in the blood of animals treated with the WT enzyme, confirming the overall advantage of the S206C mutant.

Concerning the pharmacokinetic data collected, the enzyme activity of the S206C-treated group after two hours of the single dose was significantly higher than WT- and P40S-exposed groups, likely due to less protease inactivation of the enzyme by asparaginyl endopeptidase (AEP or legumain) or cathepsin B (CTSB), as previously described [9,44,48,49,50]. Also, as related by Rathod et al. (2019), under the scheme of sensitisation, macrophage clearance of ASNase is likely to occur in the hours after the injection due to macrophage numbers increasing up to 10% every hour [24]. This result was already expected since it was reported that P40S/S206C mutations may protect and camouflage the enzyme from recognition and degradation by AEP and CTSB [9].

Previous studies indicate that even at a relatively low concentration, around 0.1 U/mL, ASNase is capable of fully depleting physiological concentrations of asparagine (Asn) within seconds in blood plasma and the central nervous system [7,51,52,53]. Here, all proteoforms were therapeutically effective for up to 12 h after injection. Also, Horvath et al. (2019) found that mice treated with three doses of 1000 U/Kg of commercial ASNase experienced a decrease in Asn concentration to half (approximately 22 µM) after 24 h of administration, and it reached its lowest level after 48 h (6 µM) [7]. In mice, the normal range of Asn is from 40 to 50 µM, as they had reported. Based on that, our scheme results are relatively close to those obtained in Horvath et al.’s (2019) experiments using the same dose as this work. It is important to highlight the central role of monocyte/macrophages (phagocytic cells) in the liver, spleen and bone marrow, which promote ASNase degradation and clearance by cathepsin B activity, thereby reducing its blood and bone marrow niche half-life. Thus, ASNase’s local rapid degradation, specifically in the bone marrow niche, mediated by CTSB, may explain a resistance mechanism of leukemic cells to escape from apoptosis [44]. Therefore, we can hypothesise a reduced affinity of S206C to surface receptors of phagocytic cells or a decrease in the lysosomal process of the proteoform and its epitope exposure, which may attenuate immune and allergic response, as observed in the S206C group data.

Finally, relative to cytotoxicity on solid tumour cells, the S206C proteoform presents lower IC_50_ values, which means a higher cytotoxic effect on Caco-2, PANC-1 and U-87 MG than WT enzyme; however, on the SK-OV-3 cell line, S206C is less effective. On the other hand, P40S has higher anti-cancer activity on U-87 MG and PANC-1, only surpassed by WT on the MDA-MB-231 cell line. Indeed, previous results from the literature described the IC_50_ against Caco-2 of 68.28 U for ASNase from *Pseudonomas aeruginosa* [54], 30 U for ASNase from *Pyrococcus furiosus* [17] and 5 U/mL from *Pyrococcus abyssi* compared to 0.41 U and 7.1 U/mL as identified here. On the other hand, cytotoxicity on PANC-1 is not well known. A study using commercial *E. coli* ASNase Spectrila^®^ identified IC_50_ of around 0.11 U/mL [55], which is more efficient when compared to 3.3 and 2.7 U/mL of our WT and S206C enzymes. Nonetheless, the S206C mutant displayed an improved immunological response compared to *E coli* ASNase (WT). Indeed, ASNase sensitivity in PANC-1 is correlated with a decrease in the purine synthesis pathway and with Gln starvation due to GLNase co-activity. Also, the resistance mechanism is likely due to glutamine synthetase gene over-expression instead of asparagine synthetase gene [55]. Regarding glioblastoma cells, a work by Karpel-Massler et al. (2016) on different glioblastoma cells treated with recombinant *E. coli* L-asparaginase from Sigma Aldrich (Saint Louis, MO, USA) effectively showed that these tumour cell lines are sensitive to ASNase, with IC_50_ values between 0.1 and 1.55 U/mL [56], while we obtained 1.2 and 0.8 U/mL for the S206C and P40S enzymes, respectively. In fact, they demonstrated the increased rates of intrinsic and extrinsic apoptosis under ASNase treatment in vitro and its enhanced inhibition of glioblastoma cells implanted into SCID SHO mice.

In conclusion, S206C mutant ASNase from *E. coli* exhibits enhanced characteristics compared to the native enzyme and recovers specific activity loss from double-mutant P40S/S206C. This novel mutant induces a lower humoral immunological response, lower allergenic and a higher protective inflammatory reaction when compared with the WT enzyme. Additionally, it demonstrates the highest enzyme activity two hours after the single dose according to the pharmacokinetic profile. Considering that macrophage-mediated recognition and clearance are higher during this period, it suggests less immune stimulation and possibly no hepatic bioaccumulation of immune complexes. Furthermore, regarding cytotoxicity on solid tumours, S206C showed a higher cytotoxic effect mainly on Caco-2, PANC-1 and U-87 MG cells than WT. Finally, we propose S206C as a promising alternative to the WT enzyme that could improve the patient’s quality of life and treatment.

Finally, as future perspectives, further specific saturation mutations in residues near or surrounding the active site [16] could be performed to develop an even better mutant enzyme with enhanced stability, activity and resistance.

## 4. Materials and Methods

### 4.1. Enzyme Production and Purification

#### 4.1.1. Gene and Vector Information

Native *E. coli* asparaginase type II enzyme (WT), encoded by *ansB* gene (UNIPROT P00805), was synthesised with optimised codon usage by GenScript (Piscataway, NJ, USA) and used as a template to obtain the following novel mutants: S206C and P40S, as well as P40S/S206C, used as a reference [9]. Briefly, the site-directed mutation was performed with the QuikChange—Site-Directed Mutagenesis Kit (Agilent Technologies, Santa Clara, CA, USA), following supplier instructions. WT and mutants were inserted into *NdeI* and *BamHI* restriction sites of the pET15b expression vector and cloned in *E. coli* BL21 (DE3) (Novagen-Merck-Millipore, Burlington, MA, USA).

#### 4.1.2. Bacterial Culture

Transformed bacteria were cultured in solid lysogenic broth medium (LB) added with carbenicillin (tryptone 10 g/L, yeast extract 5 g/L, NaCl 5 g/L, carbenicillin 50 μg/mL). Then, clones were grown in 400 mL of liquid LB and carbenicillin at 37 °C and 250 rpm overnight. Afterwards, cells from the pre-inoculum were diluted into 1 litre of fresh medium to an initial OD_600_ of 0.2, and after reaching OD_600_ of 0.7–0.8, protein production was induced by the addition of isopropyl β-D-1-thiogalactopyranoside (IPTG 1 mM) for 22 h at 37 °C. Subsequently, cells were harvested by centrifugation at 4000× *g*-force for 20 min at 4 °C and submitted to osmotic shock to obtain periplasmic fractions [9].

#### 4.1.3. Protein Purification

Osmotic shock of collected cells was performed as follows: one gram of wet cells was resuspended in 15 mL of ice-cold hyperosmotic buffer (50 mM Tris-HCl pH 8.0, 500 mM sucrose, 0.5 mM ethylenediaminetetraacetic acid-EDTA) and centrifuged at 4000× *g*-force for 30 min at 4 °C. Next, the supernatant was washed out and cells were resuspended in 0.5 mM phenylmethylsulfonyl fluoride (PMSF) and ultra-pure water (5 mL per gram of wet cell) and centrifuged at 7500× *g*-force for 20 min at 4 °C. The resultant supernatant is the periplasmic fraction, which was filtered through a 0.2 µm membrane and diluted in 50 mM Tris-HCl pH 8.8 buffer for further purification steps.

First, buffered periplasmic fraction was loaded onto a weak anion exchange chromatography column (HiTrap™ DEAE FF 5 mL, GE Healthcare Life Sciences, Chicago, IL, USA) pre-equilibrated with start buffer (50 mM Tris-HCl pH 8.8), and proteins were eluted by an increasing gradient from 0 to 100% NaCl buffer (0–500 mM NaCl in Tris-HCl 50 mM buffer, pH 8.8) in 10 column volumes (cv) at a flow of 2 mL/minute on AKTATM Start (GE Healthcare Life Sciences). All fractions containing the enzyme were pooled and concentrated on an Amicon Ultra centrifugal filter 10 kDa MWCO (Millipore Merck KGaA, Darmstadt, Germany) and submitted to size exclusion chromatography (SEC, column Superdex 200 Increase 10/300 GL, Cytiva, Uppsala, Sweden) on an AKTATM Purifier (GE Healthcare Life Sciences). ASNase was eluted using 50 mM Tris HCl and 100 mM glycine buffer, pH 7.4, with a flow rate of 1 mL/min on 5 cv. The purity of the collected fractions was estimated by SDS-PAGE (14%), stained with Coomassie Blue R-250. Finally, pure fractions were pooled and sterilised by filtration through a sterile MILLEX-GV 0.22 μm pore membrane in a laminar flow cabinet. The protein concentration of pure enzyme was determined by absorbance at 280 nm, using the molar extinction coefficient of ε = 23,505 M^−1^cm^−1^ and molecular weight (MW) of 34.5 kDa.

### 4.2. Enzymes’ Biochemical Characterization

Asparaginase and glutaminase activities were determined using the Nessler’s reagent (Merck-Millipore). The specific activity is expressed as U (1 unit corresponds to 1 μmole of ammonia released per minute) per milligram of pure protein. Briefly, the enzymes were diluted from 0.000013 to 0.00026 mg and incubated for 10 min at 37 °C with Tris HCl 50 mM buffer, pH 8.8 and L-Asn 44 mM. Likewise, for glutaminase activity, enzymes were diluted from 0.00013 to 0.0026 mg and mixed with Tris HCl 50 mM buffer, pH 8.8 and L-glutamine (Gln) 44 mM and incubated under the same conditions. Subsequently, 37 µL of the reaction volume was diluted in water with 20 µL of TCA 1.5 M to stop the enzyme activity and finally added of 37 µL of Nessler’s reagent; and the absorbance was analysed at 436 nm in a microplate reader SpectraMax M2 (Molecular Devices, San Jose, CA, USA). Blank controls were performed with all reagents except substrate (Asn and Gln) and negative controls without enzyme. Higher absorbance values were discounted from sample absorbances.

The µmole of NH3 released was calculated by interpolation of absorbance values of reactions to a standard curve of known concentrations of ammonium sulphate ((NH_4_)_2_SO_4_) vs. absorbance at 436 nm. The assay was performed in technical triplicate and analytic triplicate. Finally, the values of µmoles of NH_3_ per minute were plotted against the milligrams of the enzyme, and the slope of the linear regression curve equation corresponds to the specific activity of the enzyme (U/mg). Statistical analysis was performed with one-way ANOVA and Tukey’s multiple comparisons test on Prism 9.0 (Graphpad Software, Inc., La Jolla, CA, USA).

#### Enzymes’ Stability on Human Serum

Ten micrograms of each ASNase proteoform were incubated with sterile phosphate buffer saline (PBS) 1x, pH 7.4 and 10% of human serum (HS) (Invitrogen, Waltham, MA, USA) at 37 °C for 96 h. Control samples without HS were incubated at the same conditions. Asparaginase activity was measured at points 0, 24, 48, 72 and 96 h using Nessler’s reagent. The assay was performed with technical triplicate and analytic duplicate. Statistical analysis was performed by two-way ANOVA and Dunnett’s multiple comparisons test on Prism 9.0 (Graphpad).

### 4.3. Kinetic Parameters Determination for the Studied ASNases

The kinetic profile of all enzymes was evaluated following the coupled enzymatic reaction to glutamic-oxaloacetic transaminase (GOT G2751 Sigma, Saint Louis, MO, USA) and malic dehydrogenase (MDH M2634 Sigma, Saint Louis, MO, USA), in which for one mole of L-aspartate produced after L-asparagine hydrolysis, one equivalent mole of NADH is oxidised to NAD+, which decrease in absorbance was spectrophotometrically measured at 340 nm and 37 °C [57,58]. Thus, 80 ηM of each ASNase (WT or mutants) was mixed with Tris HCl 100 mM buffer, pH 7.4, 0.4 mM of α-ketoglutarate, 0.4 mM of NADH, 5 U/mL of GOT, 5 U/mL of MDH and with 2.5 to 600 µM of Asn for linear rates. Also, commercial asparaginase Leuginase^®^ (Beijing SL Pharmaceuticals, Beijing, China) was used for comparison. Then, the level of NADH continuous consumption in µmole was calculated based on absorbance readings at 340 nm using the Lambert–Beer law (ε = 6.22 mM^−1^ cm^−1^). The consumed µmoles of NADH were plotted against time (minutes), and the slope corresponds to the initial velocity (V_0_, µmoles of L-aspartate/minute). Kinetic parameters were determined on Prism 9 (Graphpad) using V_0_ values and Asn concentration for determining K_M_ and maximal velocity (V_max_) with non-linear regression fit. Blank and negative control reactions were performed with no substrate or enzyme addition, respectively, and the highest value was discounted of experimental absorbance. The assay was executed in technical quadruplicate and analytic duplicate.

### 4.4. Nonclinical Evaluation of ASNases

#### 4.4.1. In Vitro Assays

##### Blood Cancer Cell Lines Cytotoxic Effect of ASNases

MOLT-4 cell line was used as ALL model derived from T cells and REH as ALL pre-B cells-type. These cell lines were obtained from BCRJ (Rio de Janeiro Cell Line Bank)—Brazil. For both cell lines, 2.0 × 10^4^ cells per well were treated for 72 h with increasing ASNase doses ranging from 0.01 to 1 U/mL in clear Nunc 96-Well Flat Bottom plates (Thermo Scientific™, Rochester, NY, USA). Positive controls were performed by adding buffer, and negative controls with 20% dimethyl sulfoxide (DMSO). The cells were cultured in Advance RPMI 1640 medium supplemented with 10% of foetal bovine serum (FBS), 1% of penicillin/streptomycin 100X (P/S 1X), 0.01 M of HEPES, 1 mM pyruvate and 2.5 g/L glucose solution (Gibco Taiwan, Kaohsiung, Taiwan) at 37 °C and 5% of CO_2_. After the incubation time, 0.5 mg/mL of 3-(4,5-dimethylthiazol-2-yl)-2,5-diphenyltetrazolium bromide (MTT) (Invitrogen™, Burlington, ON, Canada) was added to each sample and further incubated for 3 h. Finally, 150 µL of a solution with 10% SDS and 0.01 M of HCl was added and incubated for 16 h to dissolve the formazan crystals, and absorbance was read at 570 nm. Inhibitory concentration (IC_50_) was calculated on Prism 9.0 (Graphpad) by Log (inhibitor) vs. normalised response, and statistical analyses of different IC_50_ within cell lines were performed by ordinary one-way ANOVA and Tukey’s multiple comparisons tests. The assay was performed with technical triplicate and analytic duplicate.

##### Solid Tumour Cell Lines Cytotoxic Effect of ASNases

Cytotoxic effect of ASNase WT and mutants was evaluated on immortalised adherent cancer cells of breast (MDA-MB-231), ovarian (SK-OV-3) (both cell lines were a gift from the lab of Professor Joy Burchell of King’s College London—currently retired), pancreas (PANC-1), colon (Caco-2) (cell lines obtained from American Type Culture Collection (ATCC)—Virginia—USA), and glioblastoma (U-87 MG) (cell line was a gift from the lab of Professor David Thurston of King’s College London—currently retired). The culture medium for MDA- MB-231 and SK-OV-3 was Advance RPMI 1640 supplemented with 10% of FBS, 1% of GlutaMAX 100× and P/S 1× (Gibco). For Caco-2, PANC-1 and U-87 MG cells, DMEM high glucose, 10% of FBS, 1% of MEM non-essential amino acids (NEAA) 100× and P/S 1× (Gibco) was used. All cell lines were cultured at 37 °C and 5% of CO_2_ at passage between 10 and 20, except for Caco-2, which was used between passages 30 and 38. In all cases, cells were seeded in clear Nunc 96-Well Flat Bottom plates (Thermo Scientific™) with 1.0 × 10^4^ cells/well and treated with 0 to 20 U/mL of each ASNase (WT, S206C and P40S) for 72 h. Following, 0.5 mg/mL of MTT was added to each well, and the plates were incubated for more than 3 h under the same conditions. Next, the media was discarded, and finally, 200 µL of DMSO was added under agitation until the crystals were dissolved. The absorbance was measured at 570 and 620 nm for reference; all assays were performed with technical and analytic triplicate. Inhibitory concentration (IC_50_) was calculated on Prism 9.0 (Graphpad) by Log (inhibitor) vs. normalised response, and statistical analyses of different IC_50_ within separated cell lines were performed by ordinary one-way ANOVA and Tukey’s multiple comparisons tests.

#### 4.4.2. In Vivo Assay

Male Balb/c SPF mice from 6- to 8-week-old were used as models in this study. Animal care was constant, changing cages twice a week, applying weight control between enzyme administrations and checking the parameters of the Grimace scale in mice, following guidance from the National Center for Replacement, Refinement & Reduction of Animals in Research NC3RS, on days of the injections. Animals had a 12 h light/12 h dark cycle, food and water ad libitum and were maintained in rooms with relative humidity at 45 to 65% and temperature 20–24 °C. In fact, they were properly monitored all over the assay, and at the beginning of each experiment, they underwent a two-week adaptation period to reduce the stress of transport and get used to the new experimentation room, where they normally fed, and no strange behaviour was ever observed.

The in vivo assay comprised three sets of independent experiments; the first one was to access toxicological effects after multiple doses, including a final challenge dose (three animals per group), as explained in Figure 7A, in which body weight, corporal temperature, blood count and general animal health conditions were evaluated. Body temperature was measured by infrared digital thermometer NX-2000 (Ningsu, Guangdong, China) for up to two hours after the enzyme injection on the administration days 0, 14 and 23 to monitor animal health and, if necessary, perform a humane endpoint. Animal weight (which was controlled throughout the entire experiment) and body temperature measurements (after each injection) were statistically analysed by two-way ANOVA, α = 0.05 and Tuckey’s multiple comparison tests on Prism 9 (Graphpad).

In the second set, the antigenic and allergenic potential of each enzyme was investigated by means of IgG, IgE, PAF and MCP-4 quantifications in plasma (five animals per group—Figure 7B). The third set of experiments aimed at the pharmacokinetics profile of the enzymes after a single dose, as shown in Figure 7C (five animals per group). ASNase doses were administered via intraperitoneal. Each set of experiments comprised four groups, namely the control group (treated only with 50 mM Tris HCl and 100 mM glycine buffer, pH 7.4), the WT group, the S206C group and the P40S group.

This project was approved by the Ethical Committee for Animal Experimentation of the School of Pharmaceutical Sciences of the Universidade de São Paulo (protocol number CEUA/FCF 036.2020-P605).

##### Organ Toxicity Analysis

After 2.5 h of the challenge dose injection of each ASNase, animals were anaesthetised with isoflurane, exsanguinated, and a blood sample was collected for blood biometry. Immediately after, they were heart perfused with 100 mL of ice-cold PBS 1x to remove all remaining blood and next fixed with 200 mL of ice-cold paraformaldehyde (PFA) 4% pH 6.9. Subsequently, various organs were harvested, rinsed with saline solution 0.9% and weighted, including the heart, liver, kidney, thymus, spleen and pancreas. Next, organs were fixed for 48 h in buffered formalin, dehydrated in graded concentrations of ethanol and embedded in paraffin, cut 3 to 5 microns thick with a microtome, and finally stained with haematoxylin-eosin (H&E) for visualisation in a light microscope by a professional unaware of the treatment. Statistical analysis of the organ’s weights was performed at Prism 9.0 (GraphPad) using Multiple *t*-tests and the two-stage linear step-up procedure of Benjamini, Krieger and Yekutieli, with Q = 1%.

##### Immunogenic and Inflammatory Potential

Blood samples were collected on days 7 and 23 in Eppendorf tubes containing EDTA 15 mg/mL, centrifuged at 8000× *g*-force, 4 °C for 3 min, and the plasma was separated and stored at −80 °C until analysis. The specific anti-ASNase IgG immunoglobulin quantification was performed by the indirect ELISA assay. Briefly, 96 well microplate F-bottom HIGH BINDING Greiner bio-one was coated with 0.6 µg/mL of asparaginase (WT or mutants) previously diluted in PBS 1x, packed in foil and incubated overnight at 4 °C. The enzyme solution (antigen solution) was removed, and the plate was washed with wash solution WS (PBS 1x with 0.5% tween 20). Then, the well bottom was sealed by adding 200 µL of PBS 1x solution supplemented with 1% of bovine serum albumin (BSA) for 90 min at 100 rpm, 37 °C and the plate packed in cling film. The wells were washed, and the plasma was diluted in work solution, PBS 1x with 10% FBS was added and incubated for 90 min at 100 rpm and 37 °C; wells were washed with WS. Next, 100 µL of secondary antibody (Anti-mouse IgG, horseradish peroxidase linked whole antibody from sheep, NA931V, GE Healthcare Life Science, Glasgow, UK) diluted 1:3000 in working solution was added and incubated for an additional 90 min, and then washed twice with WS and once with PBS 1x. After that, for the colorimetric reaction, 100 µL of 1% o-phenylenediamine (Lot#SLBT7214 Sigma, New York, NY, USA) solution in 50 mM phosphate citrate buffer, pH 5.5 and 0.03% hydrogen peroxide were added per well and the plates incubated at 37 °C under agitation for 15 min, protected from light. Finally, the reaction was stopped by adding 100 µL H_2_SO_4_ 2 M, and absorbance was read at 492 nm [58]. Specific anti-ASNase IgE (Mouse IgE ELISA kit Sigma RAB0799-1KT USA), PAF (PAF ELISA KIT CK-bio-16762, EZ Assays, Deerfield Beach, FL, USA) and MCP-4 (CCL13/MCP-4 ELISA KIT CK-bio-16613, EZ Assays, Florida-USA) quantifications were performed as the manufacturer’s instructions. Statistical analysis of IgG and IgE was performed by two-way ANOVA and Tukey’s multiple comparisons test, and PAF and MCP-4 data were analysed with an ordinary one-way ANOVA and Tukey’s multiple comparison test on Prism 9.0 (GraphPad, La Jolla, CA, USA).

##### Pharmacokinetics Profile of ASNases

As defined in Figure 7C, blood was collected in Eppendorf tubes containing EDTA at 15 mg/mL, centrifuged at 8000× *g*-force, 4 °C for 3 min, and plasma was separated and stored at −80 °C until analysis. Asparaginase activity was measured as described by Cecconello et al. (2020). Briefly, 20 µL of plasma were diluted with 180 µL of a 2 mM aspartic acid β-hydroxamate (AHA) solution dissolved in 15 mM Tris HCl buffer, pH 7.3, supplemented with 0.015% (*w*/*v*) BSA, and incubated for 30 min at 37 °C. Next, 50 µL of the resulting supernatant was added to a new tube to react with 200 µL of oxine reagent, formed by one part of 2% 8-hydroxyquinoline dissolved in absolute ethanol (*w*/*v*) and three parts of 1 M Na_2_CO_3_. Afterwards, the tube was heated at 95 °C for 1 min, cooled for 10 min and finally, absorbance was measured at 690 nm [59]. A standard curve was constructed from 0 to 2 U/mL under the same reaction conditions to interpolate the absorbance values obtained experimentally from mice plasma samples. Statistical analysis of asparaginase activity was performed at Prism 9.0 (GraphPad) using Multiple *t*-tests and the two-stage linear step-up procedure of Benjamini, Krieger and Yekutieli, with Q = 1%.

## Figures and Tables

**Figure 1 ijms-25-06008-f001:**
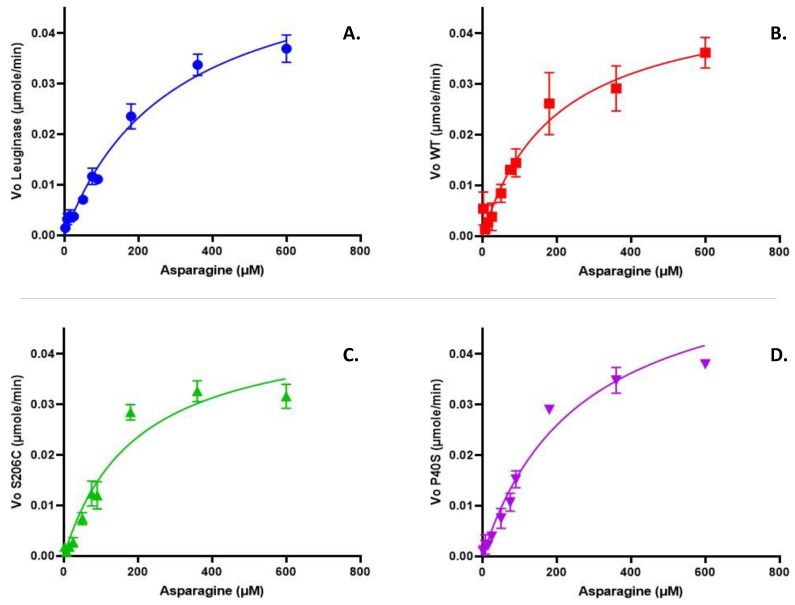
Kinetic profile of a commercial asparaginase Leuginase^®^ (**A**), and the enzymes produced in our laboratory: WT (**B**), S206C (**C**) and P40S (**D**). The µmoles of L-aspartate per minute released after asparaginase hydrolysis was quantified by the coupled enzymatic reaction, where for each mole of L-aspartate produced after L-asparagine hydrolysis, one mole of NADH is oxidised to NAD+, which decrease in absorbance was continuously measured at 340 nm and 37 °C. The analysis was performed on Prism 9 for Enzyme Kinetics—Substrate vs. Velocity with the Michaelis–Menten equation.

**Figure 2 ijms-25-06008-f002:**
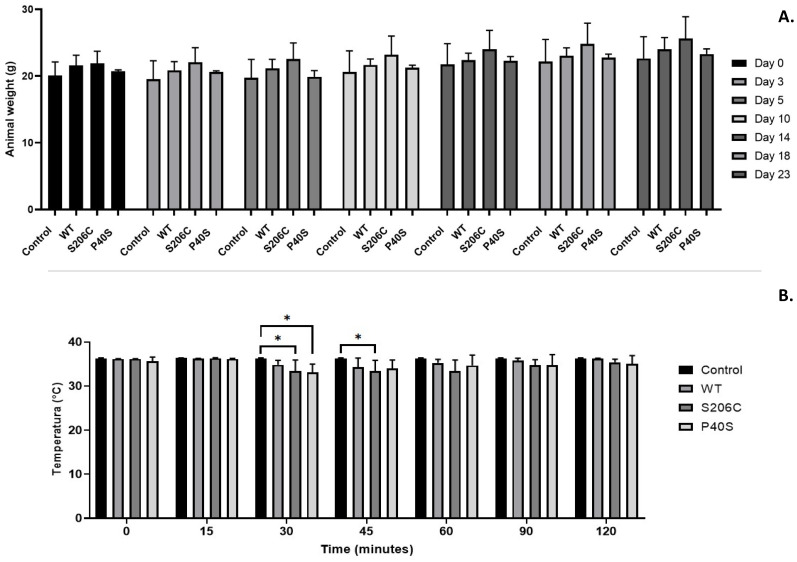
Animal health and wellness monitoring. (**A**) Weight measured on days 0, 3, 5, 10, 14, 18 and 23 of each group (Control and ASNase-treated WT, S206C and P40S) under a scheme of two doses of 1050 U/Kg on days 0 and 14 and a final dose of 5250 U/Kg on day 23. Standard deviations are shown by vertical bars, *n* = 3. No significant differences were found between groups or days of treatment. (**B**) Body temperature values for each enzyme-treated group measured for two hours after the challenge injection of 5250 U/Kg. Standard deviations are shown by vertical bars. *n* = 3. Statistical differences relative to control—*: *p*-value < 0.05.

**Figure 3 ijms-25-06008-f003:**
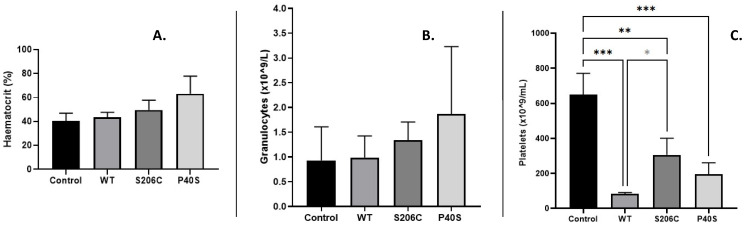
Blood biometry after challenge injection. (**A**) Haematocrit percentage, (**B**) granulocyte concentration and (**C**) platelet count for all experimental groups. Standard deviations are shown by vertical bars, *n* = 3, *: *p*-value < 0.05 (WT—S206C), **: *p*-value < 0.005, ***: *p*-value < 0.0005. No statistically significant differences were observed in haematocrit percentage or granulocytes comparing the groups.

**Figure 4 ijms-25-06008-f004:**
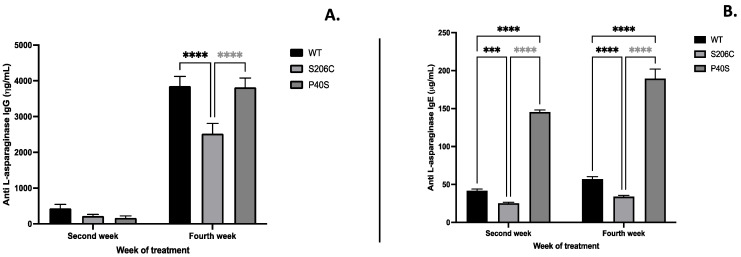
Antibody anti-ASNase quantification by ELISA of (**A**) IgG and (**B**) IgE after-treatment scheme detailed in Figure 1B. Standard deviations are shown by vertical bars, *n* = 5, ***: *p*-value < 0.0005, ****: *p*-value < 0.00005.

**Figure 5 ijms-25-06008-f005:**
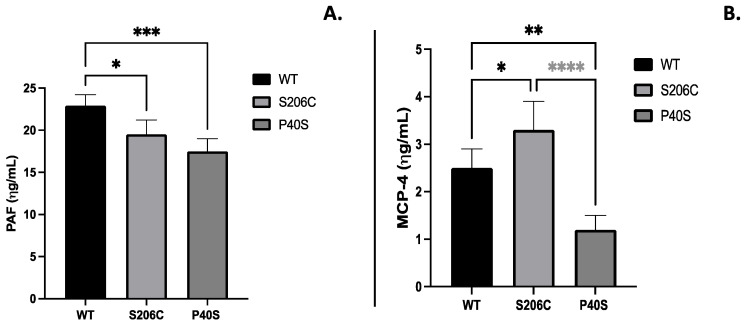
(**A**) PAF and (**B**) MCP-4 quantification by ELISA on the fourth week after treatment scheme detailed in Figure 1B. Standard deviations are shown by vertical bars, *n* = 5, *: *p*-value < 0.05, **: *p*-value < 0.005, ***: *p*-value < 0.0005, ****: *p*-value < 0.00005.

**Figure 6 ijms-25-06008-f006:**
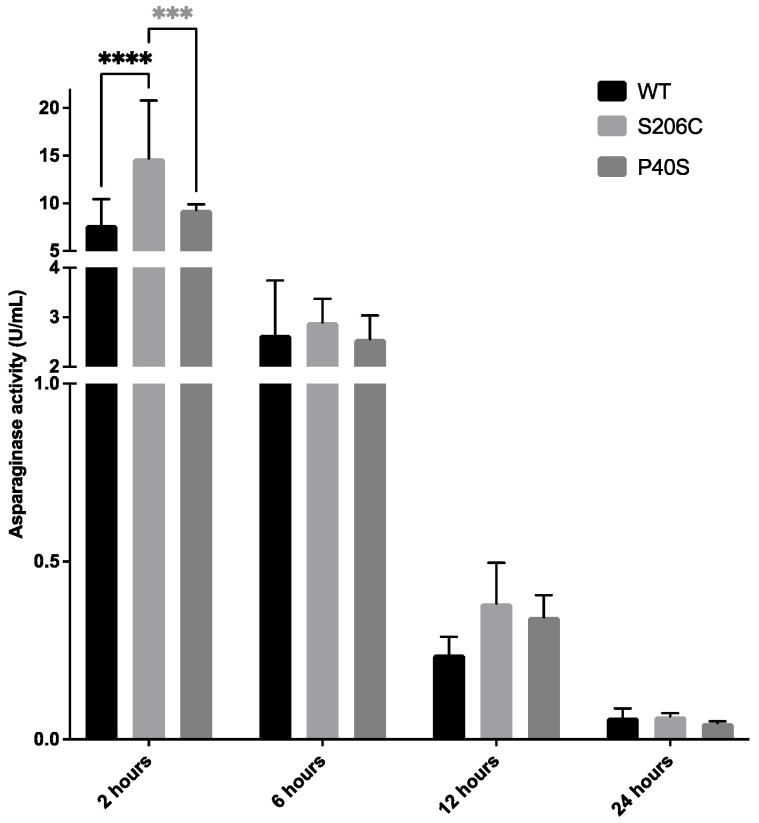
Asparaginase activity measured in mice plasma after 2, 6, 12 and 24 h of a single dose injection of 1050 U/Kg of each ASNase—WT and mutants—and quantified by oxine method. Standard deviation is shown by a vertical bar, *n* = 5, ***: *p*-value < 0.0005, ****: *p*-value < 0.00005.

**Figure 7 ijms-25-06008-f007:**
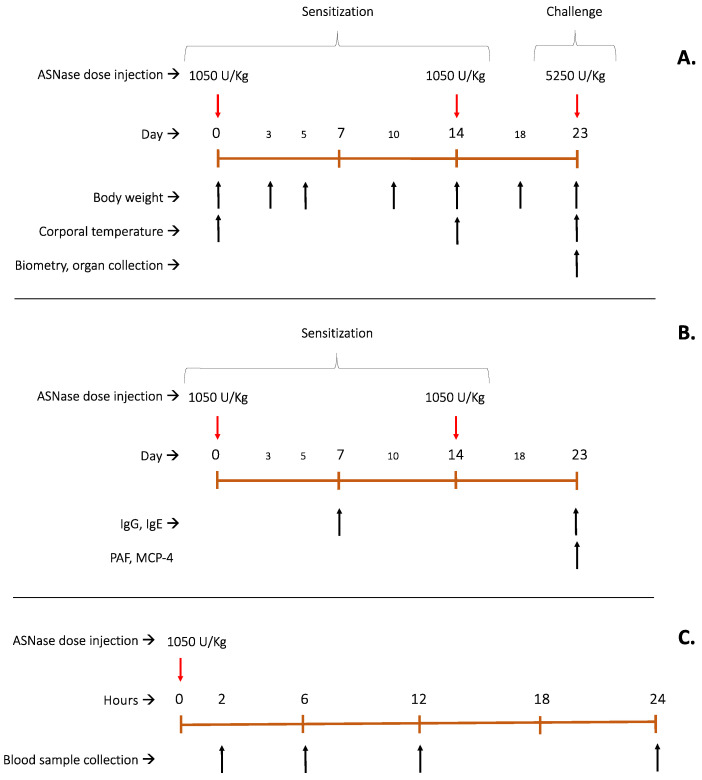
Flowchart of experimental design for in vivo studies. (**A**): Multi-organs toxicological effects of animal exposure to multiple doses of ASNase (WT and mutants) for 23 days of experimentation, with continuous weight control on days 0, 3, 5, 10, 14, 18 and 23. Temperature was measured for two hours after each ASNase injection on days 0, 14 and 23. On day 23, a blood sample was collected for biometry analysis, and organs were collected and treated with haematoxylin and eosin (H&E stain). (**B**): Antigenic and Allergenic potential evaluation of each proteoform after two sensitisation doses by plasmatic quantification of IgG and IgE (days 7 and 23) and PAF and MCP-4 (day 23). (**C**): Pharmacokinetics profile of ASNase (WT and mutants) for 24 h, with enzyme activity detection at 2, 6, 12 and 24 h after injection. Asparaginase activity was measured by the oxine method in plasma. Red arrows indicate the days of injection of ASNase; Black arrows show the days of blood collection for the analysis indicated in the figure.

**Table 1 ijms-25-06008-t001:** ASNase- and GLNase-specific activity of WT, S206C, P40S and P40S/S206C enzymes and their relative glutaminase activity percentages. Statistical differences with WT- **: *p*-value < 0.005; ***: *p*-value < 0.0005, ****: *p*-value < 0.00005. Statistical differences with S206C- ##: *p*-value < 0.005; ###: *p*-value < 0.0005, ####: *p*-value < 0.00005. Statistical differences with P40S-++: *p*-value < 0.005.

Enzyme	ASNase Activity (U/mg)	GLNase Activity (U/mg)	Relative GLNase Activity %
WT	76.22 ± 0.82	1.79 ± 0.09	2.35
S206C	74.86 ± 0.32	1.73 ± 0.04	2.31
P40S	60.82 ± 0.72 ***###	1.27 ± 0.02 **##	2.09
P40S/S206C	52.88 ± 1.12 ****####++	1.76 ± 0.05 ++	3.33

**Table 2 ijms-25-06008-t002:** Asparaginase activity of WT, S206C, P40S and P40S/S206C enzymes incubated with PBS 1x or 10% of human serum (HS) at 0 and 96 h, and the percentage of residual activity, *n* = 2. Statistical differences from 0 h to 96 h within each row—*: *p*-value < 0.05; **: *p*-value < 0.005, ***: *p*-value < 0.0005.

Enzyme	ASNase Activity (0 h)	ASNase Activity (96 h)	Residual Activity
WT-PBS 1x	88.1 ± 19.8	51.4 ± 10.1 **	58.3%
WT-HS	87.6 ± 14.8	61.7 ± 13.2	70.4%
S206C-PBS 1x	74.7 ± 5.0	43.9 ± 14.8 *	58.8%
S206C-HS	78.0 ± 3.3	64.8 ± 13.2	83.2%
P40S-PBS 1x	69.4 ± 2.5	28.0 ± 10.8 **	40.3%
P40S-HS	69.5 ± 2.0	53.1 ±15.1	76.4%
P40S/S206C-PBS 1x	61.1 ± 1.6	12.1 ± 5.6 ***	19.7%
P40S/S206C-HS	62.8 ± 1.7	27.6 ± 9.4 *	44.0%

**Table 3 ijms-25-06008-t003:** Kinetic parameters obtained for commercial Leuginase^®^, and WT, S206C and P40S ASNases. The analysis was performed on Prism 9 for Enzyme Kinetics—Substrate vs. Velocity with non-linear regression—Michaelis–Menten equation fit, *n* = 2. Statistical differences with Leuginase—*: *p*-value < 0.05. Statistical differences with S206C-#: *p*-value < 0.05.

		Leuginase	WT	S206C	P40S
Kinetic parameters	V_max_ (µmole/min)	0.05728 ± 0.0063	0.04787 ± 0.0073	0.04647 ± 0.0065	0.06001 ± 0.0055
	KM (µM)	291.3 ± 23.6	197.6 ± 17.9 *	194.0 ± 12.5 *	262.7.2 ± 9.8 #
Goodness of fit	R²	0.9744	0.9203	0.9362	0.9389

**Table 4 ijms-25-06008-t004:** IC_50_ (U/mL) of WT, S206C and P40S ASNases for different blood cancer cell lines (MTT assay) after 72 h of treatment. *n* = 2, Statistical differences with WT—*: *p*-value < 0.05. No significant differences were observed between mutants.

Cell Line	ASNase Proteoform		
	WT	S206C	P40S
MOLT-4	0.112 ± 0.020	0.076 ± 0.012	0.045 ± 0.013 *
REH	0.114 ± 0.019	0.149 ± 0.031	0.114 ± 0.002

**Table 5 ijms-25-06008-t005:** IC_50_ (U/mL) of WT, S206C and P40S ASNases for different solid cancer cell lines (MTT assay) treated for 72 h. *n* = 3. Statistical differences with WT—*: *p*-value < 0.05; **: *p*-value < 0.005; ***: *p*-value < 0.0005. Statistical differences with S206C-#: *p*-value < 0.05; ##: *p*-value < 0.005.

Cell Line	ASNase Proteoform		
	WT	S206C	P40S
MDA-MB-231	5.9 ± 0.075	6.3 ± 0.434	7.5 ± 0.584 **#
CACO-2	9.2 ± 0.522	7.1± 0.249 *	9.0 ± 1.074 #
SK-OV-3	8.5 ± 1.344	11.9 ± 0.43 *	9.4 ± 1.013
PANC-1	3.3 ± 0.166	2.7 ± 0.075 **	2.9 ± 0.201 *
U-87 MG	1.5 ± 0.107	1.2 ± 0.099 *	0.8 ± 0.098 ***##

**Table 6 ijms-25-06008-t006:** The average weight of organs collected after 23 days of treatment with multiple doses of ASNase (groups WT and mutants) and control group, *n* = 3. The statistical difference with the control group—*: *p*-value < 0.05. No statistical significance was identified between WT and mutants.

Organ	Weight (mg)			
	Control	WT	S206C	P40S
Heart	147.1 ± 7.3	162.8 ± 4.0	151.9 ± 3.7	153.8 ± 11.5
Spleen	99.1 ± 18.0	100.2 ± 2.3	94.3 ± 10.3	91.7 ± 3.2
Liver	1529.2 ± 38.2	1846.7 * ± 50.1	1686.7 ± 163.7	1827.0 * ± 34.7
Kidney	317.5 ± 27.0	291.9 ± 1.1	264.8 ± 7.2	251.5 ± 12.5
Pancreas	940.0 ± 2.9	774.8 * ± 21.3	732.2 * ± 30.2	713.3 * ± 38.1
Thymus	85.6 ± 13.9	99.4 ± 11.7	57.6 ± 2.0	84.7 ± 7.0

## Data Availability

All data supporting the results can be found within the manuscript.

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
