# Peer review of "Nonclinical Evaluation of Single-Mutant E. coli Asparaginases Obtained by Double-Mutant Deconvolution: Improving Toxicological, Immune and Inflammatory Responses"

_ijms, 2024, doi:10.3390/ijms25116008_

Round 1

Reviewer 1 Report

Comments and Suggestions for Authors

G. Monteiro and coworkers have continued their previous work (ref 9) concerning protein engineering of asparaginase with the aim of obtaining mutants ensuring high antibody production (in mice). In the earlier work, they applied the well-known mutational technique of error-prone PCR, and obtained the double mutant P40S/S206C. Unfortunately, its activity was greatly reduced.

In the present work, the double mutant was deconvoluted with formation of the two respective single mujtants P40S and S206C, which is a clever strategy. The single mutant P40S was foun d to cause the trouble, because it lowered activity significantly. In contrast, the single mutant S206C alone led to a 2-fold increase in activity relative to wildtype.

This work lives up to the high standard of Int. J. Molec. Sci. Due to the potential significance in human pharmaceutical application, this reviewer recommends publication following minor revision:

1. When citing protein engineering methods, the authors should also cite some recent reviews in this research area, e.g., H. Zhao et al, Chem. Rev.2021, 121, 12384-12444; G. Qu et al, Angew. Chem. Int. Ed. 2020, 59, 13204-13231.

2. On the basis of their previous (and present) work, it seems that rather than using random mutagenesis, focused saturation mutagenesis at residues surrounding or near the active site (see again:  G. Qu, Angew. Chem. Int. Ed. 2020, 59, 13204-13231) may well lead to better results. The authors should state this in the conclusion section as a perspective.

3. The authors present an excellent example of gaining mechanistic insight in an enzyme by the deconvolution of their double mutant. Indeed, the high value of deconvolution of multi-mutational variants as a useful procedure and strategy has been delineated a decade ago (M. T. Reetz, Angew. Chem. Int. Ed. 2013, 52, 2658-2666). The authors should mention this concept at the appropriate place in the text.

Comments on the Quality of English Language

Only minor style problems.

Author Response

We would like to thank you for the opportunity to submit a revised version of our manuscript to the International Journal of Molecular Sciences. We greatly appreciate the time and effort that you have dedicated to contributing to the improvement of this work.

The suggestions mentioned by the reviewers were properly analysed, carried out and highlighted as requested. Also, at cover letter was composed to adequately explain our responses to the referees’ comments.

REVIEWER 1

First comment

1. When citing protein engineering methods, the authors should also cite some recent reviews in this research area, e.g., H. Zhao et al, Chem. Rev.2021, 121, 12384-12444; G. Qu et al, Angew. Chem. Int. Ed. 2020, 59, 13204-13231.”

Response

We would like to thank you for your positive feedback. About the requested reference, we incorporated your suggestion and believe those references enhance information on our reference 15, which cites protein bioengineering specifically in relation to asparaginase.

Second comment

2. On the basis of their previous (and present) work, it seems that rather than using random mutagenesis, focused saturation mutagenesis at residues surrounding or near the active site (see again:  G. Qu, Angew. Chem. Int. Ed. 2020, 59, 13204-13231) may well lead to better results. The authors should state this in the conclusion section as a perspective.”

Response

We agree with this comment and have, accordingly, incorporated the suggestion.

Third comment

3. The authors present an excellent example of gaining mechanistic insight in an enzyme by the deconvolution of their double mutant. Indeed, the high value of deconvolution of multi-mutational variants as a useful procedure and strategy has been delineated a decade ago (M. T. Reetz, Angew. Chem. Int. Ed. 2013, 52, 2658-2666). The authors should mention this concept at the appropriate place in the text.”

Response

We agree with this comment and have, accordingly, incorporated the suggestion.

Reviewer 2 Report

Comments and Suggestions for Authors

Ruiz-Lara et al. reported the two mutants of L-asparaginase (P40S and S206C) and evaluated the toxicogical, immune and inflammatory responses to asparaginase in vitro and in vivo. This work was based on their previous mutant P40S/S206C. The results showed that these two mutants have low toxicity and high specific enzyme activity. The detailed comments were listed as below:

1) The authors have published a similar article in Biochemical Pharmacology previously. In the present work, they evaluated the single mutant and compared with the double mutant. In this regard, the title should be changed that protein engineering could be removed.

2) The data in Figure 1 and Table 1 was repreated. Thus remove figure 1 or table 1.

3) To evalute the specific activity of the mutants, it is better to dock the mutant and wild type enzyme with substrate to explain the results.

4)To investigate the stability of mutants, molecular dynamics could be performed.

5) Some citations, line 340, 433-434, line 470 should be inserted and add the corresponding references.

6) Add more structrual content in the discussion part.

Comments on the Quality of English Language

The English is fine.

Author Response

We would like to thank you for the opportunity to submit a revised version of our manuscript to the International Journal of Molecular Sciences. We greatly appreciate the time and effort that you have dedicated to contributing to the improvement of this work.

The suggestions mentioned by the reviewers were properly analysed, carried out and highlighted as requested. Also, cover letter was composed to adequately explain our responses to the referees’ comments.

REVIEWER 2

First Comment

1) The authors have published a similar article in Biochemical Pharmacology previously. In the present work, they evaluated the single mutant and compared with the double mutant. In this regard, the title should be changed that protein engineering could be removed.”

Response

We appreciate your careful review and detailed suggestions. We agree with this comment and have, accordingly, modified the title.

Second Comment

2) The data in Figure 1 and Table 1 was repeated. Thus remove figure 1 or table 1.”

Response

We agree with this comment and have moved Figure 1 to supplementary material (now Figure S3).

Third and fourth Comment                                                     

3) To evalute the specific activity of the mutants, it is better to dock the mutant and wild type enzyme with substrate to explain the results.

4)To investigate the stability of mutants, molecular dynamics could be performed.”

Response

The specific activity of the mutants were experimentally measured (Table 1) and in addition, kinetic parameters were determined (Table 3). We observed that P40S loss in activity was caused by diminishing of enzyme affinity for asparagine substrate. In relation to mutants’ stability, we presented experimental quantification of residual activities followed during 96 hours (Table 2). Molecular dynamics were performed and discussed on the previous work by Rodrigues et al (2020), specifically as shown in figure 3 of that publication, which also contains all information about the double mutant P40S/S206C (reference 9 in our work).

Fifth Comment

5) Some citations, line 340, 433-434, line 470 should be inserted and add the corresponding references.”

 Response

We appreciate these important observations and we have already highlighted the citations.

Sixth Comment

6) Add more structrual content in the discussion part.”

Response

We thank the effort of the reviewer to improve the quality of our manuscript. However, in this specific point, the aim of the present data is not to evaluate the biochemical and structural features of asparaginase. We want to show that it is necessary more options of asparaginase versions, because we believe that personalized medicine is the future, as stated in the last paragraph of the introduction:

“…Our results indicate that mutation P40S was detrimental to enzyme activity and isolated S206C mutant rescued WT ability to hydrolyse asparagine. Nonetheless, both single mutants presented activity against ALL cell line. Interestingly, single mutants presented different behaviours related to activity against solid tumour cell lines, induction of antibody titres and resulted in specific physiologic alterations related to biopharmaceutical toxicity. Overall, the S206C mutant seems to be the best option, but each proteoform contains different features and can be more adequate for different situations and/or patients, reinforcing the importance of generating options of ASNase versions for future personalized therapeutic application.”

Round 2

Reviewer 2 Report

Comments and Suggestions for Authors

The authors have addressed the corresponding issues in the revised manuscript. It could be accepted after minor modifications and the detailed comments are listed below:

1) "Biobetter development" is inappropriate, so please revise the title.

2) There are some grammar errors. For example, lines 87-89, correct the tense.

Comments on the Quality of English Language

There are some grammar errors.

Author Response

We really appreciate your suggestions and attempted to present the improved version of the manuscript.

REVIEWER 2

First comment

1) "Biobetter development" is inappropriate, so please revise the title.

Response

We would like to thank you for your positive feedback. We modified to reflect exactly what we performed “Nonclinical evaluation of single mutant E. coli asparaginases obtained by double-mutant deconvolution: improving toxicological, immune and inflammatory responses”

Second comment

2) There are some grammar errors. For example, lines 87-89, correct the tense.”

Response

We agree with this comment and have, accordingly amended it. As well as some other grammar or spelling errors all over the text.
